# GENERALIZED GRAPH EMBEDDING MODELS

## ABSTRACT

Many types of relations in physical, biological, social and information systems can be modeled as homogeneous or heterogeneous concept graphs. Hence, learning from and with graph embeddings has drawn a great deal of research interest recently, but only *ad hoc* solutions have been obtained this far. In this paper, we conjecture that the one-shot supervised learning mechanism is a bottleneck in improving the performance of the graph embedding learning algorithms, and propose to extend this by introducing a multi-shot unsupervised learning framework. Empirical results on several real-world data set show that the proposed model consistently and significantly outperforms existing state-of-the-art approaches on knowledge base completion and graph based multi-label classification tasks.

## 1 INTRODUCTION

Recent studies have highlighted the importance of learning distributed representations for symbolic data in a wide variety of artificial intelligence tasks (Bengio et al., 2013). Research on word embeddings (Mikolov et al., 2013) has led to breakthroughs in many related areas, such as machine translation (Bahdanau et al., 2015), question answering (Xiong et al., 2016), and visual-semantic alignments (Karpathy & Fei-Fei, 2017). However, learning to predict for large-scale knowledge graphs (KGs) is still a challenging problem left, this is largely due to the *diversity* of the ontologies, and the *semantic richness* of the concepts, which makes it really hard to generate proper and universally applicable graph embeddings, simply based on word-level embeddings (Cai et al., 2017).

Being able to generate reasonable and accurate distributed representations for large-scale knowledge graphs would be particularly valuable, in that it may help predict unobserved facts from limited concepts, uncover gaps in our knowledge, suggest new downstream applications, which clearly reflects the central concerns of the artificial intelligence (Nickel et al., 2016a; Henaff et al., 2017). Therefore, massive attention has been devoted to the potential of embedding entities and relationships of multi-relational data in low-dimensional vector spaces in recent years (Wang et al., 2017).

In this paper, we consider the problem of developing simple and efficient model for learning neural representation of **generalized knowledge graphs**, including the multi-relational *heterogeneous* graphs, and more specifically defined *homogeneous* graphs (such as social and biological networks).

Following the pioneer work of Nickel et al. (2011) and Bordes et al. (2013), almost all of the state-of-the-art approaches try to model the graph embedding learning problem as supervised binary classification problems, their objective functions are usually *one-shot* (single purpose) . We argue that prior research in this area might have been affected and biased by " established priors", which prevents the formulation of a methodology that is objective enough to cope with the highly sparse knowledge graphs. We propose to handle the embedded learning problem of knowledge graphs with an unsupervised neural network model, called the *Graph Embedding Network* (**GEN**). The proposed model consists of three simple multi-layer perceptron (MLP) cells, each cell operates in response to a different "query" with regard to the input fact, which will be trained sequentially. The formulation of the model is inspired by the neural sequence-to-sequence (seq2seq) model (Sutskever et al., 2014), except that we attempt to use the MLP cells to mimic the sequence learning capability of the recurrent neural network (RNN), to model the semantic structure of the knowledge graphs.

The major contribution of this paper is that: (1) we propose GEN, a novel and efficient multi-shot framework for embedding learning in generalized knowledge graphs. (2) We show how GEN is in accordance with established principles in cognitive science, providing flexibility in learning representations that works on graphs conforming to different domains.

## 2 RELATED WORKS

During the last few years, an increasing amount of research attention has been devoted to the challenge of representation learning on knowledge graphs, especially focused on the potential benefits for the knowledge base completion (KBC) tasks, including the *link prediction* problem and the *relation prediction* problem. Among which, the relation translating model TransE (Bordes et al., 2013), the tensor factorization based semantic matching model RESCAL (Nickel et al., 2011), and the neural network based semantic matching model ER-MLP (Dong et al., 2014; Nickel et al., 2016b), are probably the most heavily studied from the methodology perspective. For good surveys on such embedding learning algorithms, see Nickel et al. (2016a), Wang et al. (2017), and Cai et al. (2017).

Broadly speaking, related works can be divided into two categories: linear and non-linear, according to whether the output embedding has a reasonable linear interpretation. State-of-the-art linear models include the TransE, RESCAL, TranH (Wang et al., 2014), DistMult (Yang et al.), and ANALOGY (Liu et al., 2017), while the popular non-linear models include the ER-MLP, ComplEX[1] (Trouillon et al., 2016), HoIE (Nickel et al., 2016b), ProjE (Shi & Weninger, 2017) and ConvE (Dettmers et al., 2017). The proposed GEN model is also a non-linear model.

The graph embedding learning models that is most closely related to this work is probably the ProjE model, which makes use of an embedding projection function defined as:

$$\mathbf{h}(\mathbf{r}, \mathbf{t}) = g(\mathbf{w}_0 \cdot f(\mathbf{w}_1^r \mathbf{r} + \mathbf{w}_1^t \mathbf{t} + \mathbf{b}_1) + \mathbf{b}_0)$$

where $\mathbf{h}, \mathbf{r}, \mathbf{t}$ denote the embedding vectors, $f(\cdot)$ and $g(\cdot)$ are non-linear activation functions, $\mathbf{w}_0$, $\mathbf{w}_1^r$ and $\mathbf{w}_1^t$ are learnable weight matrices, $\mathbf{b}_0$ and $\mathbf{b}_1$ are bias vectors. The output ranking scores of entity $h$ with regard to the given query $(?, r, t)$ can be obtained through a softmax function:

$$Score(h_i, r, t) = softmax\left\{\mathbf{h}(\mathbf{r}, \mathbf{t})\right\}_i$$

However, as one could see from above functions, the ProjE model is built upon the query $(?, r, t)$, hence is a one-shot solution, which is distinctly different from our GEN model. Still another difference lies in the definition of the objective loss function, the ProjE model choose to use the (selective) cross-entropy loss based on the open world assumption, while our model uses a simplified cross-entropy loss based on the close world assumption. In order to save the computation cost, the ProjE model introduced a negative sampling process, this could cause potential risks for introducing additional bias. Besides, its candidate sampling process is time consuming and hard to be paralleled.

Another model that is closely related to the GEN model is the ER-MLP model, which can be interpreted as creating representation for each element of triples and deriving their existence from this representation (Nickel et al., 2016a). The ER-MLP model can be defined as:

$$Score(\mathbf{h}, \mathbf{r}, \mathbf{t}) = \mathbf{w}^T g\left\{\mathbf{C}^T(\mathbf{h} \oplus \mathbf{r} \oplus \mathbf{t})\right\}$$

where symbol $\oplus$ denotes the *vector concatenation* operator, vector $\mathbf{w}$ and matrix $\mathbf{C}$ are global weight vectors shared by all the entities and relations, $g(\cdot)$ is an element-wise non-linear activation function. This model is built upon the fourth query as we defined in Section 3, it is a supervised solution, which is quite different from ours. One well-known disadvantage of the ER-MLP is that, even properly regularized, it is still easily prone to over-fitting on knowledge graph datasets (Nickel et al., 2016b), therefore we do not compare with it in this work, but instead with the ProjE model.

As mentioned before, the primary motivation of this study is to develop a graph embedding model that is universally applicable to a wide variety of situations. In order to verify the validity of our solution on heterogeneous networks, we further test it on *multi-label network classification* tasks for social networks (BlogCatalog) and biological networks (Protein-Protein Interaction), and compare our results with two state-of-the-art techniques, namely, DeepWalk (Perozzi et al., 2014) and node2vec (Grover & Leskovec, 2016). Both of them are derived directly from the word2vec model (Mikolov et al., 2013), which creating node embeddings of the graphs based on the skip-gram framework, and train the model with *corpus* generated through random walking on that graph. However, it is shown that the random walk sampling can be insufficient for supervised learning tasks in the sparse network environment (Liu et al., 2016). Our results support this conjecture, the experimental results on benchmark tests provide strong evidence that our model performs much better.

---

[1]The ComplEX model can be seen as an extension of the DistMult model in the complex space, albeit there is no nonlinear transformations applied, we treat it as a non-linear model here.

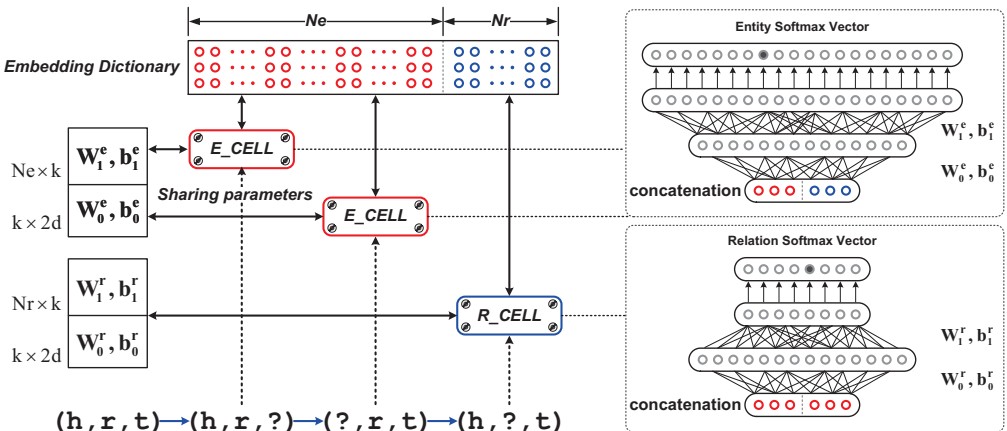

Figure 1: GEN: A Graph Embedding Model

# 3 APPROACH AND MODEL ARCHITECTURE

Most of the prevalent semantic knowledge databases are built upon the Resource Description Framework (RDF) , in which the *facts* are represented and stored in the form of SPO (Subject, Predicate, Object) ***triples***. Following the convention, we will use the symbol $(h, r, t)$ to represent a unit of *facts*, in which $h, r$ and $t$ denote the *head* entity, the *relation*, and the *tail* entity, respectively.

The primary motivation of this paper is to develop a representation learning method that is suitable and flexible enough for modeling different types of knowledge graphs from a universal perspective. To achieve this objective, the most important problems to be faced are associated with: *how to define the optimization problem and how to solve it*. As mentioned above, previous works only consider a **one-shot** mapping from the embedding space to the criterion space, which we conjecture, would be vulnerable to loss considerable amount of the structured semantic information. For instance, if given a fact (*Elvis Presley, profession, singer*), one could immediately learn the following queries:

- Q1: What is the *profession* of *Elvis Presley*? A1: *singer*.
- Q2: Can you name a person whose *profession* is *singer*? A2: *Elvis Presley*.
- Q3: What is the possible relationship in between *Elvis Presley* and *singer*? A3: *profession*.
- Q4: Is it true that *Elvis Presley*'s *profession* is *singer*? A4: Yes.

In fact, this is the actual way we humans learn the meaning of concepts expressed by a statement. These self-labeled queries reflect the following modeling philosophy: (1) $(h, r) \Rightarrow t$; (2) $(t, r) \Rightarrow h$; (3) $(h, t) \Rightarrow r$; (4) $(h, r, t) \Rightarrow$ T/F; respectively. This has been exclusively adopted by the previous research. However, none of them have systematically investigated the effect of combination all of such information. In this section, we propose a novel **multi-shot** model to solve this problem. For a more detailed discussion of the motivation and intuition behind this model, see Appendix A.

## 3.1 OVERVIEW OF THE MULTI-SHOT LEARNING FRAMEWORK

The proposed model (GEN) is designed to process data in sequential form. As shown in Fig.1, GEN consists of three components (cells), each corresponding to an individual query with regard to the given input triple. In this study, we propose to use a 2-layer MLP network to deal with the parameter estimation problem for each query individually, although it can be substituted by any other one-shot models, we only report the test results on MLP cells for simplicity. In training mode, the training set is fed into the system sequentially, each of the triple is decomposed into three self-labeled queries: $(h, r, ?) \Rightarrow t$, $(?, r, t) \Rightarrow h$, and $(h, ?, t) \Rightarrow r$. Each of the queries is fetched into the corresponding cell in order to update the parameters. Since for any given triple, our model would read it from three different perspective, we call it "multi-shot model" to distinguish it from other related works.

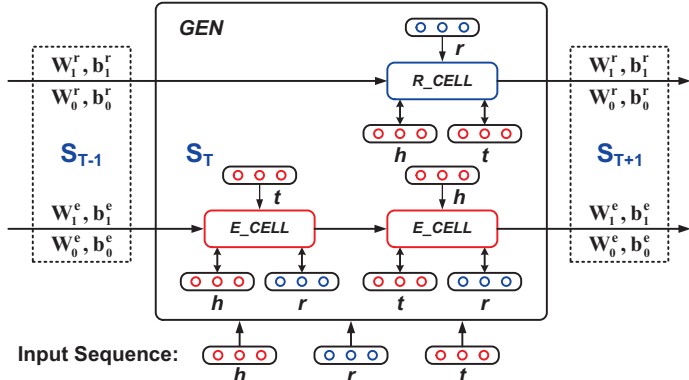

Figure 2: The seq2seq representation of the GEN model.

Parameters of the model can be logically divided into two parts. Firstly, the distribution representation of the *entities* and the *relations* are defined in the same $d$-dimensional space, which, as shown in Fig.1, are organized together as a *learnable* dictionary of embeddings. Secondly, there exist two types of MLP cells in the model, one deals with the entity prediction tasks, the other is responsible for the relation prediction tasks, which are marked as "E_CELL" and "R_CELL" respectively. Each individual cell has its own parameter set $\{\mathbf{W}_0, \mathbf{b}_0; \mathbf{W}_1, \mathbf{b}_1\}$ representing certain network structures. Please note that two E_CELLs are introduced to learn from the labeled entities, based on query $(h, r, ?)$ and $(?, r, t)$. According to our modeling hypothesis, which claims that all of the *relations* should be treated conceptually instead of syntactically, we propose to share parameters between the E_CELLs, the intuition behind is to let them share their *memory* of each known *facts* from both side of the *relation*, so that after training with enough knowledge, the E_CELLs will eventually able to learn how to correctly distinguish between valid and invalid entities for the given queries.

Another theoretical explanation of the GEN model is given below. We consider the proposed model as a variant of the RNN model, or more precisely, a neural seq2seq model, as illustrated in Fig.2. When training with the graph (a "document" of triples), the GEN model is organized as a stacked RNN, which consists of two chains: the E_CELL chains and the R_CELL chains. For any given input $(h, r, t)$, each of the cells works as an individual seq2seq model according to its responsive query. For instance, the R_CELL is responsible to query $(h, ?, t) \Rightarrow r$, it will take the embedding of $h$ and $r$ as input, and take $r$ as its target label, and the parameters (memory) of the R_CELL will be updated through back-propagation according to the discrepancy between the prediction results (in this case the softmax vector) and the desired label $r$. Therefore, the proposed model is completely unsupervised, which is distinctly different from previous works. Also please note that due to the lack of semantic connections between the adjacent triples in the input sequence, we did not consider the "long term memory" in this model, as usually did in *real* RNN models. Therefore, there only exists one "global memory" in this model — the parameter of the two types of cells, which is responsible for "learning to remember" the rules of how the knowledge graph is constructed.

## 3.2 DEFINITION OF THE GEN CELLS

The network structure of the E_CELLs and the R_CELLs are quite similar, the only difference is that they have different number of neurons in the hidden layer and the output layer, which are defined as hyper-parameters as shown in Fig.1. For simplicity, we only present the implementation details of the E_CELLs here. In order to answer query $(h, ?, t) \Rightarrow r$, the hidden layer of the E_CELL takes input from the embedding dictionary according to label $h$ and $r$, the hidden layer is defined as:

$$\mathbf{x}_1 = f(\mathbf{W}_o^e \cdot \mathbf{x}_0 + \mathbf{b}_0) \tag{1}$$

where $\mathbf{x}_0 = [\mathbf{h} \oplus \mathbf{r}]$, denotes the concatenation of the embedding vectors, hence the $\mathbf{x}_0$ is a $2d \times 1$ real-value vector. $\mathbf{W}_o^e$ is a $k \times 2d$ weights matrix, $\mathbf{b}_0$ is a $k \times 1$ bias vector, $k$ denotes the number of neurons in the hidden layer, and $f(\cdot)$ is a non-linear activation function, in this work, we use the

rectified linear unit (ReLU) function for all the experiments (Nair & Hinton, 2010). The output layer takes the hidden state vector $\mathbf{x}_1$ as input, mapping it to the target label space:

$$\hat{\mathbf{y}} = g(\mathbf{W}_1^e \cdot \mathbf{x}_1 + \mathbf{b}_1) \tag{2}$$

where $\mathbf{W}_1^e$ is a $N_e \times k$ weights matrix, $\mathbf{b}_1$ is a $N_e \times 1$ bias vector, $N_e$ denotes the number of entities in the dictionary, $g(\cdot)$ denotes the softmax function. Hence, $\hat{\mathbf{y}}$ is a $N_e \times 1$ probability vector, which means that, when training the model with a given fact $(h, r, t)$ to answer the query $(h, r, ?)$, the predictive results output by the model is a probabilistic distribution over all of the possible candidate entities. The cross-entropy loss with regard to prediction results is then defined as:

$$\mathcal{L}(\hat{\mathbf{y}}) = -\sum_{i=1}^{N_e} \mathbf{y}[i]log(\hat{\mathbf{y}}[i]) + (1 - \mathbf{y}[i])log(1 - \hat{\mathbf{y}}[i]) \tag{3}$$

where $\mathbf{y}$ denotes the ground truth, which is a one-hot vector exclusively activated by $t$. To speed-up the stochastic convex optimization process, we use a mini-batch setting, and rewrite the averaged cross-entropy loss over a batch of multiple samples of size $N$ as following simplified form:

$$\mathcal{L}(\mathbf{y}) = -\frac{1}{N} \sum_{i=1}^{N} log(\hat{\mathbf{y}}_i[\, t_i \,]) \tag{4}$$

where the subscript $i$ denotes the $i$-th sample of the batch, $t_i$ represent the index of label $t$ in the ground truth vector of that sample. Eq.4 is computationally efficient, however, it tend to ignores the existing knowledge for query $(h, r, ?)$ other than the current fact $(h, r, t)$, which has been proven to be useful for improving performance (Shi & Weninger, 2017). But, our experimental results show that the impact of such a problem can be controlled by means of collaborative correction with related facts under our model framework, which further demonstrate the validity of our modelling assumptions. Hopefully, the lessons learned for designing reasonable and computationally efficient cost functions in this study can serve as exemplars for future work.

## 4 EXPERIMENTAL RESULTS

We evaluate the proposed model on two distinctly different types of graph embedding learning tasks. Firstly, we evaluate our model on knowledge base completion tasks with the conventional datasets FB15K and WN18[2], and their upgrade version FB15k-237 and WN18RR[3]. Secondly, we evaluate our model on graph based multi-label classification tasks with two benchmark datasets from the complex network research area: BlogCatalog and Protein-Protein Interaction (PPI)[4]. Background information of the datasets and the implementation details of our model are given in Appendix B.

### 4.1 EVALUATION ON KNOWLEDGE BASE COMPLETION TASKS

The aim of the first evaluation was to assess the performance of the proposed model in link prediction tasks, by comparing it with other state-of-the-art approaches. We report the *filtered P@N* scores following the protocols proposed by Bordes et al. (2013), which means that all of the known facts would be screened out from the ranking list before calculating the statistics of the *hits*. The numerical results are presented in Table 1, where the highest scores in each column are presented in bold.

We reproduced all of the results of the existing studies (mostly with the released code), whereas some of which are below the reported record. For a fair comparison of the models, we cite those numbers from the original publications (marked with ⋆ symbols). Also, it seems that results reported by Dettmers et al. (2017) only consider the *tail* entity prediction scenario (without averaging with the *head* entity prediction results), hence we report two version of the test results of our model, the averaged version is named as GEN(avg.), while the *tail* entity prediction results are reported with

---

[2] Available online at: https://everest.hds.utc.fr/doku.php?id=en:transe
[3] Available online at: https://github.com/TimDettmers/ConvE
[4] Available online at: https://snap.stanford.edu/node2vec/

Table 1: Link prediction results on WN18, FB15K and WN18RR, FB15K-237 (symbols: $\star$ denotes the value is cited from the original source, $\dagger$ denotes the result comes from (Dettmers et al., 2017))

| Datasets | WN18 | | | WN18RR | | | FB15K | | | FB15K-237 | | |
|---|---|---|---|---|---|---|---|---|---|---|---|---|
| Measures | P@1 | P@3 | P@10 | P@1 | P@3 | P@10 | P@1 | P@3 | P@10 | P@1 | P@3 | P@10 |
| TransE | 44.5 | 85.9 | 93.8 | 2.7 | 33.1 | 42.7 | 36.1 | 59.0 | 76.2 | 17.6 | 29.6 | 44.6 |
| TransH | 33.7 | 79.3 | 87.4 | 1.9 | 33.7 | 40.4 | 33.0 | 59.1 | 70.7 | 19.3 | 34.0 | 44.7 |
| HoIE | 93.0$^\star$ | 94.5$^\star$ | 94.9$^\star$ | 35.6 | 37.8 | 39.3 | 40.2$^\star$ | 61.3$^\star$ | 73.9$^\star$ | 8.2 | 15.2 | 26.1 |
| Analogy | **93.9**$^\star$ | 94.4$^\star$ | 94.7$^\star$ | 37.9 | 39.2 | 41.0 | 64.6$^\star$ | 78.5$^\star$ | 85.4$^\star$ | 13.2 | 22.8 | 37.2 |
| DistMult | 72.8$^\star$ | 91.4$^\star$ | 93.6$^\star$ | 38.9$^\dagger$ | 43.9$^\dagger$ | 49.1$^\dagger$ | 54.6$^\star$ | 73.3$^\star$ | 82.4$^\star$ | 15.5$^\dagger$ | 26.3$^\dagger$ | 41.9$^\dagger$ |
| ComplEX | 93.6$^\star$ | 94.5$^\star$ | 94.7$^\star$ | **41.1**$^\dagger$ | **45.8**$^\dagger$ | **50.7**$^\dagger$ | 59.9$^\star$ | 75.9$^\star$ | 84.0$^\star$ | 15.2$^\dagger$ | 26.3$^\dagger$ | 41.9$^\dagger$ |
| ConvE | 93.5$^\dagger$ | **94.7**$^\dagger$ | **95.5**$^\dagger$ | 30.6$^\dagger$ | 36.0$^\dagger$ | 41.1$^\dagger$ | 67.0$^\dagger$ | 80.1$^\dagger$ | 87.3$^\dagger$ | 22.0$^\dagger$ | 33.0$^\dagger$ | 45.8$^\dagger$ |
| ProjE | 75.7 | 87.8 | 95.1 | 31.8 | 41.7 | 46.0 | 57.5 | 66.32 | 88.4$^\star$ | 17.3 | 28.0 | 43.0 |
| GEN(avg.) | 64.2 | 91.8 | 94.1 | 37.8 | 40.2 | 43.0 | 76.4 | 84.1 | 88.8 | 20.4 | 31.3 | 45.8 |
| GEN(opt) | 90.6 | 94.1 | 94.5 | 38.3 | 40.5 | 43.1 | 77.7 | 84.7 | 89.0 | 20.8 | 32.1 | 46.2 |
| GEN(tail) | 65.0 | 91.8 | 94.2 | 39.0 | 41.7 | 44.5 | **78.9** | **86.9** | **91.6** | **29.5** | **42.3** | **57.7** |

model named GEN(tail). Besides, we found that our model tends to remember the reverse facts with regard to the triples that has been processed during the training phase. We argue that this is an inherent characteristic of our modeling methodology, since it would treat such reverse facts as *conceptually correct*. Therefore, we also report P@N scores after screening out such reverse facts, this model is named as GEN(opt). We consider that under certain practical circumstances, it is reasonable to care about such results, because the *reverse facts* are direct reflections of the known facts, and in many scenarios, they themselves are useful and effective facts.

From Table 1 one could see that the performance of ComplEX seems much more competitive than other models on both of the WordNet subset, however, according to our tests, TransE and HoIE perform (generalized) more stable than others for all of the subtasks. Also please note that, after filtering out the *reverse facts* from the ranking list, we recorded a significant increase in P@1 score on WN18, which was not observed in other models. Since most of the semantic relations defined in WordNet are reflexive (Miller, 1995), we believe that these results help verify the efficacy of our model framework. Further evidence can be found by looking at evaluation results on FB15K and FB15K-237, in which our model consistently and significantly outperforms others for all settings.

The goal of the second evaluation was three-folded. (1) To assess the relation prediction performance of our model. (2) To verify the validity of the multi-shot learning framework. (3) To evaluate the quality (representability) of different embedding schemes. To achieve this goal, we carried out a group of experiments depicted in Table 2, where the model name shown in the parentheses indicate that the test is based on the embeddings generated by that model, but being re-trained with our model for fair comparison. For example, before testing the GEN(TransE) model, we need to train a GEN model with TransE embeddings, the only difference is that the pre-trained embeddings will not be updated during the training process, such that the quality of the different embedding schemes can be assessed more objectively. The results of GEN(HoIE) were obtained similarly from the pre-trained HoIE embeddings. The pre-trained word2vec embedding[5] and GloVe embedding[6] are obtained from the publicly available dictionaries released respectively by Google and Stanford NLP Group for research purpose, which are also heavily studied by recent researches. For entities and relations consisting of many words, we use the weighted sum of the word embeddings as their distributed representation for the test. The three models listed in the bottom of Table 2 demonstrate the one-shot learning capability of GEN, for instance, the results of GEN($h, r \Rightarrow t$) were obtained by only considering the query $(h, r, ?)$ during the training stage.

From the studies, the following conclusions could be obtained. (1) The performance of GEN on relation prediction tasks has been demonstrated. However, it seems that such strong performance mainly comes from our GEN framework, under which the predictive capability of a variety of em-

---

[5]Available at: https://code.google.com/archive/p/word2vec; version: GoogleNews-vectors-negative300.

[6]Available at: https://nlp.stanford.edu/projects/glove/; file version: glove.42B.300d.

Table 2: Empirical comparison of the embedding schemes on FB15K dataset

| Tasks | Head Entity Prediction | | Tail Entity Prediction | | Relation Prediction | |
|---|---|---|---|---|---|---|
| Models \ Measures | P@1 | P@10 | P@1 | P@10 | P@1 | P@10 |
| GEN(GloVe) | 39.79 | 68.80 | 44.64 | 74.72 | 85.24 | 98.57 |
| GEN(word2vec) | 48.05 | 75.81 | 52.09 | 81.34 | 86.50 | 98.77 |
| GEN(HoIE) | 30.55 | 58.86 | 35.66 | 64.84 | 92.28 | 99.68 |
| GEN(TransE) | 47.91 | 77.58 | 52.25 | 82.75 | 93.15 | 99.71 |
| GEN | **73.85** | **86.01** | **78.86** | **91.64** | 93.99 | **99.75** |
| GEN($h, r \Rightarrow t$) | 36.18 | 62.88 | 36.85 | 63.38 | 86.61 | 98.49 |
| GEN($t, r \Rightarrow h$) | 32.47 | 58.11 | 40.40 | 67.72 | 86.44 | 98.41 |
| GEN($h, t \Rightarrow r$) | 26.34 | 49.42 | 30.11 | 54.41 | **94.11** | 99.75 |

beddings can be enhanced. In considering the ratio of the number of facts to relations involved, this problem seems much easier than the link prediction problem. (2) The validity of the multi-shot framework has been verified, since each of the one-shot GEN model performs significantly worse than the multi-shot model for almost all the tests, except that in relation prediction tasks, GEN($h, t \Rightarrow r$) performs comparable to GEN, this is probably because that it was exclusively trained for that task, which is prone to overfit the data. (3) Comparing with their performance on link prediction tasks, we argue that the embeddings generated by GEN are probably more representative and informative than other embedding schemes, which we will provide more empirical (visual) evidence in Appendix C.

## 4.2 EVALUATION ON GRAPH BASED MULTI-LABEL CLASSIFICATION TASKS

In previous section, the term "knowledge graph" was used to refer to a multi-relational database, in which the entities were engaged in one or more heterogeneous relations, which means the relations related with a entity may range over different domains. In this section, we consider the problem of embedding learning on another type of graph — the homogeneous graphs (networks), in which the entities were engaged in a specific relationship, which is a natural structure people use to model the physical world, such as the various social network and the biological information systems. In this study, we consider it as a generalized form of the knowledge graphs, and attempt to come up with a general-purpose framework that could be used for embedding learning on different graphs.

To verify the validity of the proposed model, we evaluate GEN by comparing its performance on some benchmark multi-label classification tasks with the state-of-the-art DeepWalk and Node2vec models. Besides, we also report results on TransE and HoIE embeddings for comparison purpose, the supervised model used for multi-label classification are identical to each other (but differ from the embeddings). For fair comparison, all of the results with regard to the DeepWalk (Perozzi et al., 2014) and Node2vec (Grover & Leskovec, 2016) are cited from their original sources.

Following the convention of previous authors, we randomly sample a portion of the labeled nodes as training set (and the rest are used for test), we repeat this process 9 times (with the training ratio increased from 10% to 90%), and report two of the averaged measures (w.r.t. recall, precision, and F1-measure) on each of the test, namely, macro-average and micro-average. The Macro-F1 weights equally all the *categories* regardless of how many *labels* belong to it, while the Micro-F1 weights equally all the *labels*, thus favouring the performance on common *categories*.

Numerical results are presented in Table 3 and 4 respectively, the highest scores in each column are presented in bold face. From Table 3 one could see that the performance of DeepWalk proves much more competitive than other models when labeled data is sparse, but GEN still consistently outperforms when given 50% of the data, which demonstrates the validity of the proposed embedding learning framework for modeling author connections on social networks. Next, we investigate the performance of our model on even more sparse graphs, i.e. the Protein-Protein Interactions network. Table 4 shows that GEN performs consistently and significantly better than other baselines. In fact, when trained with only 20% of the labeled proteins, GEN performs significantly better than other approaches when they are given 90% of the data. We argue that this strong performance not only indicates that our model is flexible enough to the biological networks, but also provides new insights into their underlying biological mechanisms. Also please note that Macro-F1 scores in Table 3

Table 3: Multi-label classification results on BlogCatalog dataset

| Measures | Models | 10% | 20% | 30% | 40% | 50% | 60% | 70% | 80% | 90% |
|---|---|---|---|---|---|---|---|---|---|---|
| Micro-F1 | DeepWalk | **36.00** | **38.20** | **39.60** | **40.30** | 41.00 | 41.30 | 41.50 | 41.50 | 42.00 |
| | Node2vec | 34.64 | 36.15 | 36.63 | 37.01 | 37.20 | 37.38 | 38.05 | 38.27 | 40.91 |
| | TransE | 16.71 | 17.10 | 17.44 | 17.64 | 17.77 | 18.50 | 19.13 | 19.62 | 20.50 |
| | HoIE | 30.88 | 33.31 | 34.63 | 35.70 | 36.17 | 37.31 | 40.21 | 38.79 | 40.69 |
| | GEN | 27.61 | 31.38 | 35.02 | 38.55 | **41.19** | **44.40** | **45.78** | **48.87** | **51.84** |
| Macro-F1 | DeepWalk | **21.30** | **23.80** | 25.30 | 26.30 | 27.30 | 27.60 | 27.90 | 28.20 | 28.90 |
| | Node2vec | 16.52 | 18.81 | 19.81 | 20.09 | 20.97 | 21.50 | 22.37 | 23.16 | 24.60 |
| | TransE | 2.69 | 3.09 | 3.33 | 3.52 | 3.41 | 3.85 | 4.14 | 4.63 | 5.33 |
| | HoIE | 13.86 | 17.10 | 18.98 | 20.84 | 20.77 | 22.65 | 25.64 | 23.06 | 27.79 |
| | GEN | 19.32 | 23.26 | **26.74** | **31.06** | **33.53** | **36.57** | **38.83** | **40.27** | **44.60** |

Table 4: Multi-label classification results on PPI dataset

| Measures | Models | 10% | 20% | 30% | 40% | 50% | 60% | 70% | 80% | 90% |
|---|---|---|---|---|---|---|---|---|---|---|
| Micro-F1 | DeepWalk | 15.36 | 17.40 | 18.26 | 19.41 | 19.75 | 20.23 | 20.46 | 21.52 | 21.79 |
| | Node2vec | 16.32 | 17.94 | 19.14 | 19.68 | 20.32 | 21.80 | 21.76 | 22.50 | 22.88 |
| | TransE | 12.80 | 17.69 | 20.94 | 23.57 | 24.58 | 27.32 | 30.42 | 31.84 | 35.20 |
| | HoIE | 14.85 | 18.95 | 21.52 | 24.58 | 27.55 | 29.34 | 31.03 | 33.56 | 35.71 |
| | GEN | **16.36** | **27.31** | **27.97** | **32.73** | **38.10** | **42.85** | **46.43** | **51.09** | **55.16** |
| Macro-F1 | DeepWalk | 12.93 | 14.46 | 15.94 | 17.05 | 17.74 | 18.05 | 18.41 | 18.52 | 20.03 |
| | Node2vec | 13.00 | 15.56 | 16.82 | 17.28 | 17.92 | 18.37 | 19.60 | 20.72 | 21.28 |
| | TransE | 8.71 | 11.45 | 16.43 | 19.00 | 20.37 | 22.69 | 25.42 | 27.35 | 30.53 |
| | HoIE | 9.36 | 16.10 | 17.55 | 20.76 | 23.96 | 24.92 | 26.82 | 30.26 | 32.45 |
| | GEN | **14.74** | **25.83** | **27.04** | **31.27** | **35.98** | **40.82** | **45.02** | **50.35** | **52.92** |

and 4 demonstrate that, comparing with other embedding schemes, GEN performs more stable (and better) in both common and rare categories, which indicates that the embeddings generated by GEN are probably more representative and informative than other solutions, thus the supervised model built on top of it is less vulnerable to global under-fitting and local over-fitting.

## 5 CONCLUSION AND FUTURE WORK

Representation learning of knowledge graphs is a key concern for artificial intelligence and cognitive science. Many types of relations in physical, biological, social and information systems can be modeled with concept (knowledge) graphs. In this paper, we present an efficient scalable framework for learning conceptual embeddings of entities and relations in generalized knowledge graphs, including the homogeneous and heterogeneous graphs. We give evidence that the proposed model learns good representations of all these graphs for knowledge inference and supervised learning. For future work, we plan to investigate more thoroughly the efficacy of the proposed modeling framework, with respect to the decomposition of the semantic information conveyed by the linked concepts into elementary information, i.e. the four Q&A pairs. Also, we seek to enhance the quality of scientific investigations and theoretical conceptualizations on graph embedding learning in the context of *semantic interoperability*, for there is usually no possibility to interpret the *embedded* information meaningfully and accurately in order to produce useful results as defined by existing algorithms.

ACKNOWLEDGMENTS

We are grateful to the anonymous reviewers for taking time read and provide helpful comments.

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

APPENDIX A: MOTIVATION AND INTUITION

To get an intuitive understanding of the problem, consider the following examples taken from three typical KGs that have been heavily studied by the academic and industrial communities:

- (*Elvis Presley, instance of, rock star*) : taken from the *WordNet*[7], one of the largest online lexical database of English, in which each distinct concept (called *synset*) are interlinked by means of rigidly defined (hence limited) conceptual-semantic or lexical relations.

- (*Elvis Presley, /people/person/profession, singer*) : taken from the *Freebase*[8], which was once to be the largest collaboratively edited knowledge base (deprecated at this time and absorbed by the Wikidata project). In which each named entities are interlinked by means of fine-grained relation types defined in the meta-schema. Due to the loosely-defined nature of the relation types, redundancy or alternate facts are allowed to exist simultaneously, such as, (*Elvis Presley, profession, musician*) and (*Elvis Presley, profession, actor*).

- (*Elvis Presley, rdf:type, American rock singers*) : taken from the *YAGO*[9], one of the largest and most active semantic knowledge base developed at the Max Planck Institute for Computer Science in Saarbrücken , which combines the clean taxonomy (relation types) of WordNet with the richness of the Wikipedia category system (classes of entities ).

As can be perceived from above examples, the use of different ontologies can lead to different (and incoherent) relations between the same pair of concepts, similarly, applying different ontologies can lead to diverse kinds of conceptualizations. Therefore, it is (arguably) impractical to rely on using the word-level embeddings to precisely represent the knowledge graphs under the diverse conditions, and it is necessary to develop a *universal* solution that is applicable to all of the ontology infrastructures, for **phrase-level** embedding learning of the different concept representations.

As mentioned in Section 3, in order to develop a representation learning method that is flexible enough for modeling different types of knowledge graphs, the most important problems to be faced are associated with how to define the optimization problem and how to solve it. According to our survey, most state-of-the-art models, including the translating models derived from the TransE (Bordes et al., 2013; Lin et al., 2015), the latent semantic models derived from the RESCAL (Nickel et al., 2011; 2016b), and the neural network models derived from the NTN (Socher et al., 2013), were all trying to define the graph embedding learning problem as a *supervised binary classification* problem, in which the optimization objectives are defined in the form of a *relation-specific* cost function of the entity and/or relation embeddings, and then to solve it with a stochastic gradient decent (SGD) process. Typical criteria used to evaluate the cost functions include the *logistic loss* and the *pairwise margin-based criterion*, and the negative samples used for training the model are usually sampled from the complement of the knowledge graph based on the open world assumption (Drumond et al., 2012). However, we doubt that there are many situations where such modeling strategies would have theoretical and practical disadvantages.

Firstly, we speculate that the reason why most previous studies did not consider the first and second queries simultaneously (see Section 3), is probably due to the difficult in modeling the inverse semantic relatedness of the entities from the given fact. In other words, shall we use the embedding of $r$ to represent its reverse $r'$? If we do so, it seems that it will inevitably lead to semantic paradox like: *Presley*'s *profession* is *Presley*, since from the model's perspective, there is no difference between the entity *Presley* and other entities that may appear on both side of the relation *profession*. Considering the sparsity of the knowledge graph, models trained with limited facts would very likely tend to give higher scores to the entities that have been "seen in the right place".

In order to solve this problem, we propose **to model the facts conceptually** instead of concretely (or literally, syntactically), which means that we will focus on the semantic meanings of the embeddings (of the entities and relations), rather than their syntactic features. Such a conceptual embedding scheme allow us to unify the representation of a relation (r) and its reverse counterpart (r'), and to accommodate the lexical variety in use by various knowledge bases.

---

[7]http://wordnet.princeton.edu
[8]https://developers.google.com/freebase
[9]https://www.mpi-inf.mpg.de/departments/databases-and-information-systems/research/yago-naga/yago

Table 5: Statistics of the data sets

| Dataset | # entities | # relations | # training set | # validation set | # test set |
|---------|-----------|-------------|----------------|------------------|------------|
| WN18 | 40,943 | 18 | 141,442 | 5,000 | 5,000 |
| WN18RR | 40,943 | 11 | 86,835 | 3,034 | 3,134 |
| FB15K | 14,951 | 1,345 | 483,142 | 50,000 | 59,071 |
| FB15K-327 | 14,541 | 237 | 272,115 | 17,535 | 20,466 |

| Dataset | # nodes | # edges | # categories | # labels | Avg. labels |
|---------|---------|---------|--------------|----------|-------------|
| BlogCatalog | 10,312 | 333,983 | 39 | 14,476 | 1.40 |
| PPI | 3,890 | 38,292 | 50 | 6,640 | 1.70 |

The intuition behind is, for any given fact $(h, r, t)$, one would instantly recognize the bidirectional semantic connection between $h$ and $t$, without need of translating it into $(t, r', h)$ explicitly in his/her mind. We believe this is crucial for efficient utilization of the structure information of the KGs for representation learning, empirical evidence is provided in Section 4 and Appendix C, respectively.

Secondly, we propose to use unsupervised learning techniques for graph embedding learning tasks, because: (1) Presently, almost all of the large-scale knowledge graphs are extremely sparse, which would unavoidably degrade the quality and reliability of the supervised learning algorithms. Further, considering the relation-specific solution that dominates the current research, the situation might get even worse. (2) Selecting negative examples for pair-wise training would be tricky and expensive, since in practice, it is very hard to generate a "proper and informative" negative sample responsive to each of the positive examples. For example, when learning from the fact (*Einstein, employer, IAS*), the false fact (*Einstein, employer, Stanford*) would seem to be more reasonable and informative than (*Einstein, employer, singer*) — if the objective is to further improve the predictive capability of the model to discriminate between similar objects.

To solve the *data sparsity* problem, we propose to model each of the facts as a short sentence, the entire KG can be regarded as a huge document, so that it can be processed by unsupervised encoder-decoder neural models, which has demonstrate to be efficient and useful in concept learning from the large-scale and feature sparse data (Sutskever et al., 2014). In order to avoid the sampling bias due to the selection of uninformative entities, we propose to use the *softmax cross-entropy loss* as a measure of the predictive discrepancy for model training, because its probability interpretation is more objective than those squared or logistic errors conventionally used in this area, and, it has been proven to be convex for the MLP we used in this paper (Bengio et al., 2005).

## APPENDIX B: BACKGROUND INFORMATION AND IMPLEMENTATION DETAILS

### B.1 DATASETS

WN18 is a subset of WordNet, which contains mostly the conceptual-semantic and lexical relations, and the entities are organized in a strictly hierarchical manner. FB15k is a subset of Freebase, which contains facts gathered from Wikipedia, mostly focused on the topic of movies and sports.

These datasets have been used as a de facto benchmarks for comparative evaluation, however, recent research (Toutanova & Chen, 2015; Dettmers et al., 2017) show that the test sets of WN18 and FB15k contain a lot of reversed triples that have been presented in the training set, i.e., $(h, r, t)$ versus $(t, r, h)$. Which, we consider would favor our model over those one-shot alternatives.

Therefore, we provide results on FB15k-237, which is introduced by Toutanova & Chen (2015), it is a subset of FB15K where reversing relations are removed. And, we also test on WN18RR provided by Dettmers et al. (2017), which is a reverse duplication free new sample of WordNet.

The multi-relational data sampled from WordNet and Freebase can be seen as typical of the heterogeneous graphs, in order to verify the generality of the developed model, we also perform evaluation in the multi-label classification setting on some typical homogeneous graphs.

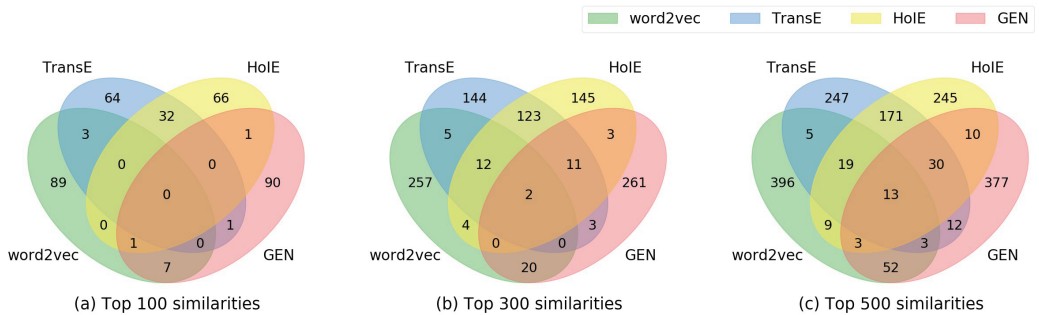

Figure 3: Venn diagrams of the top-N cosine similarity ranking results of the relation embeddings generated by four typical embedding schemes on FB15K dataset.

BlogCatalog is a social network sampled from the BlogCatalog website, which contains only one relationship: the social connection between the blog authors, while the labels represent the interested topic categories provided by the bloggers. Protein-Protein Interactions is a biological network sampled from the PPI network for Homo Sapiens, which also contains only one relationship: the existence of interactions between the proteins, while the labels represent the biological states of the proteins. In the training set of these graph corpus, every entity (node) is assigned one or more labels from a finite set, the task is to predict the labels for the nodes in the test set.

The statistics of these data sets are summarized in Table 5.

## B.2 Experimental Setup

We optimized the hyper-parameters of all the datasets via extensive grid search and selected the model with the best filtered P@10 score on the validation set. Hyper-parameter ranges for the grid search were the following: embedding dimension $d$ in $\{50, 100, 200, 300\}$, hidden layer dimension $k$ in $\{256, 512, 1024, 2048\}$ , MLP dropout rate $p$ in $\{0.0, 0.1, 0.2, 0.3\}$, learning rate $\eta$ in $\{0.001, 0.01, 0.1, 1, 5, 10\}$, learning rate decay $\lambda$ in $\{0.7, 0.75, 0.8, 0.85, 0.9, 0.95\}$. In this study, we use the following combination of parameters for all of the graph embedding learning tasks :

- E_CELLS: $\{d : 200, k : 2048, p : 0.2, \eta : 5, \lambda : 0.9$.
- R_CELLS: $\{d : 200, k : 512, p : 0.2, \eta : 5, \lambda : 0.9\}$.
- Mini-batch Settings: $\{\text{batch\_size} : 512, \text{epoch} : 50\}$

For multi-label classification tasks, we implement a single layer perceptron model for multi-task learning with: $\{k : 128, \eta : 0.1, \lambda : 0.9\}$, which is selected through grid search with the best averaged Macro-F1 score on randomly sampled validation set from the labeled nodes.

## Appendix C: Investigating and Visualizing the Embedding Schemes

In this section, we provide qualitative analysis on four typical embedding schemes (GEN, HoIE, TransE and word2vec), with the intention of better understanding the connection between the existing graph embedding schemes, and highlighting areas that remain poorly understood for further investigation. The reason we choose these models (except the word2vec) is because, according to our tests, they have demonstrated to be efficient and scalable to large-scale problems, and are also exhibiting good generalization ability on real data sets. We also consider the word2vec embeddings because we found that with the help of our multi-shot learning model, it achieves state-of-the-art performance on most of the knowledge base completion tasks (see Section 4), which is interesting, and worth some consideration (probably indicates a promising potential for transfer learning).

The argument in the first case is to claim that the graph embeddings generated by GEN are different from other solutions. We calculate the cosine similarities for each pair of the 1,345 relations in

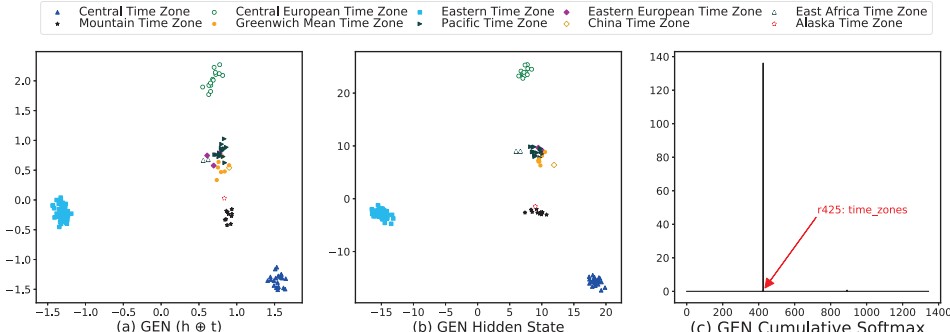

Figure 4: Visualization analysis of the GEN embedding space by using of the principal component analysis on embedding of the entities for relation prediction tasks. The case is taken from the FB15K test set, with all of the triples related to relation #425: /location/location/time_zones.

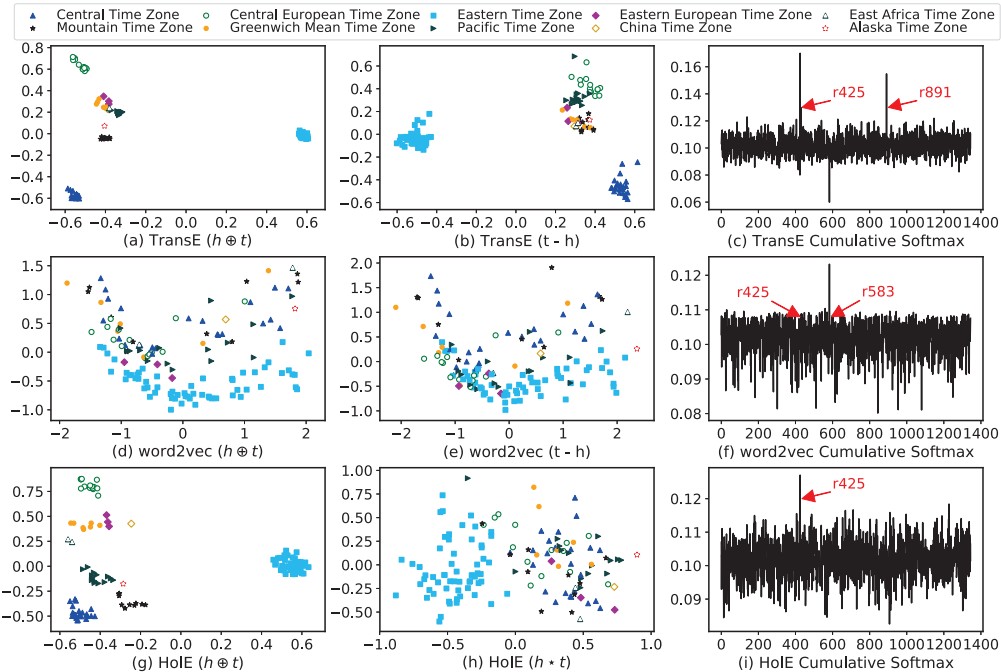

Figure 5: Visualization analysis of the TransE, HoIE and word2vec embedding schems by using of the principal component analysis on embedding of the entities for relation prediction tasks. The case is taken from the FB15K test set, relation #425: /location/location/time_zones.

FB15K w.r.t. four embedding schemes respectively, and compare their top-N ranking list (which is a set of relation pairs for each embedding scheme) through Venn diagrams, as illustrated in Fig.3.

Figure 3 reveals very different clustering pattern between GEN and other alternatives in their corresponding embedding space. What's particularly interesting is that, although the TranE model is inspired by and heavily reliant on the spatial distribution of the wrod2vec embeddings (Bordes et al., 2013), but they are, in fact, not similar at all. On the contrary, the results of TransE and HoIE share a lot of similarities, it appears that in their top-300 and top-500 lists, almost half of the relation pairs were contained in their intersection. Which probably indicates that the translating embedding hypothesis (Bordes et al., 2013) is theoretically similar in nature to the holographic embedding hypothesis (Nickel et al., 2016b), when used for graph modeling. This is not an easily testable hypothesis, we consider it to be an open question, which we hope to further explore in the future.

The goal of the second experiment is to verify the claim that the embeddings generated by GEN are more representative and informative than other embedding schemes. Here we provide a case study on a randomly selected relation from FB15K, namely "/location/location/time_zones". There are 137 triples related to this relation (#425) in the test set, all of the head entities are names of the countries or regions, and the tail entities are the corresponding time zones. The heads are uniquely different from each other, while there are only 10 different *time zones* existed in the tails.

We plot all of the 137 triples in Fig.4, in which (Fig.4a and Fig4b) the *input* multi-dimensional vectors are projected to a 2-dimensional subspace spanned by $x$ and $y$, by using of the principal component analysis (PCA), then we choose the first two principal components as the principal axes. In Fig.4a, the *input* is the concatenation of the head and tail entity of each triple, i.e. $(h \oplus t)$, with the intention of investigating the patterns of such *feature vectors* for relation prediction tasks. Hence, we choose the name of the tails as legend labels. As can be seen from Fig.4a, the *feature vectors* of the 137 triples shows clear clustering tendencies with regard to the categories in their tail entities. Based on this observation, we further plot the hidden layer of the R_CELL (which is a 512-dimensional vector in this case) located before the output layer in our GEN model, as depicted in Fig.4b. From Fig.4b one could see that the distance between the data points is amplified, and the distinction becomes more prominent. We plot the cumulative softmax in Fig.4c, in which the X-axis represents the 1,345 type of relations in FB15K, Y-axis denotes the cumulative softmax values. The curve is obtained by adding all of the softmax vectors output by GEN with regard to the 137 triples. Obviously, the only peak observed in Fig.4c clearly exhibit that GEN can make good use of these (concatenated) features to identify the corresponding relations correctly.

For comparison purpose, we also visualize the other three embedding schemes with the same protocol, as illustrated in Fig.5. Since the corresponding models do not use MLP for relation prediction, we can not plot their "hidden state" and "accumulate softmax" for the second and the third subplots, hence we choose to visualize their predictive criterion vectors and output ranking list instead. The processing practice is consistent with the protocol of the original literature. Specifically, for TransE, we plot $(t - h)$ as the *hidden state* for relation prediction, and calculate the $\ell 1$-norm distance $|r_i - (t - h)|_1$ w.r.t each of the relation $r_i$ in FB15K, then we process the distance vector with the softmax function for calculation of the *accumulate softmax*. While for HoIE, we plot the circular correlation vector $(h \star t)$ as the *hidden state*, and calculate the cosine similarity of $(h \star t) \cdot r_i$ w.r.t each of the relation $r_i$ in FB15K, then we use the obtained (cosine) vector to calculate the *accumulate softmax*. For word2vec embeddings, we use the same protocol as dealing with TransE.

From Fig.5 one could see that, the concatenated embedding vectors of TranE and HoIE shows similar clustering pattern as the GEN case, which help explaining the reason that under our multi-shot learning framework, the embeddings generated by these models perform similar in relation prediction tasks (see Table 2). It also provides evidence for our conjecture that these two embedding schemes could be inherently similar to each other. Form their criterion vectors (the second subplot for each models), one could see that their clustering pattern is not as clear as the case of GEN, which help explain their performance on relation prediction tasks (as shown in the third subplot)[10]. We consider this as solid support for the validity of the proposed multi-shot learning framework.

More evidence could be found with our source code release, which will be made publicly available on GitHub to encourage reproducible research, after anonymous review.

---

[10] The alternative peaks appeared in subplot Fig.5c and Fig.5f are:

- #891: "/base/schemastaging/phone_open_times/time_zone", and
- #583: "/time/time_zone/locations_in_this_time_zone".

