# OpenReview forum: "Generalized Graph Embedding Models"
_ICLR.cc/2018/Conference — Reject_

### Official Review · AnonReviewer3 · 2017-11-27
**The paper is about a generalized knowledge graph embedding approach which learns the embeddings based on three different simultaneous objectives: predicting the head, tail or relation based on the information from the other two.**

**Rating:** 6
**Confidence:** 4

**Review:**

The paper is well-written and provides sufficient background on the knowledge graph tasks. The current state-of-the-art models are mentioned and the approach is evaluated against them. The proposed model is rather simple so it is really surprising that the proposed model performs on par or even outperforms existing state-of-the art approaches.


? The E_CELLs share the parameters. So, there is a forced symmetry on the relation i.e. given input head h and relation r predicting x and given input relation r and tail t predicting y would result in the same entity embedding x=y with h=t?

? In Table 2, you report the results of the retrained models GEN(x). There, the weights for the MLPs are learned based on the existing embeddings which do not get changed. I am missing a comparison of the change in the prediction score. Was it always better than the original model? Did all models improve in a similar fashion?

? Did you try training the other models e.g. TransE with alternating objective functions for respectively predicting the head, tail or relation based on the information from the other two?

? Are the last 3 Gen(x,y -> z) rows in Table 2 simple MLPs for the three different tasks and not the parts from the overall joint learned GEN model?

? Why is a binary classifier for Q4 not part of the model?

? Is the code with the parameter settings online?


+ outperforms previous approaches

+ proposes a general use case framework

- no run-time evaluation although it is crucial when one deals with large-scale knowledge graphs


Further comments:
* p.4: “it will take the embedding of h and r as input, and take r as its target label” -> “it will take the embedding of h and t as input, and take r as its target label”
* “ComplEX” -> “ComplEx”

---

### Official Review · AnonReviewer1 · 2017-11-27

**Rating:** 4
**Confidence:** 4

**Review:**

This paper tackles the task of learning embeddings of multi-relational graphs using a neural network. As much of previous work, the proposed architecture works on triples (h, r, t) wth h, t entities and r the relation type.


Despite interesting experimental results, I find that the paper carries too many imprecisions as is.
* One of the main originality of the approach is to be able for a given input triple to train by sequentially removing in turn the head h, then the tail t and finally the relation r. (called multi-shot in the paper). However, most (if not all) approaches learning embeddings of multi-relational graphs also create multiple examples given a triple. And that, at least since "Learning Structured Embeddings of Knowledge Bases" by Bordes et al. 2011 that was predicting h and t (not r). The only difference is that here it is done sequentially while most methods sample one case each time. Not really meaningful or at least not proved meaningful here.
* The sequential/RNN-like structure is unclear and it is hard to see how it relates to the data.
* Writing that the proposed method "unsupervised, which is distinctly different from previous works" is not true or should be rephrased. The only difference comes from that the prediction function (softmax and not ranking for instance) and the loss used.  But none of the methods compared in the experiments use more information than GEN (the original graph). GEN is not the only model using a softmax by the way.
* The fact of predicting indistinctly a fact or its reverse seems rather worrying to me. Predicting that "John is_father_of Paul" or that "John is_child_of Paul" is not the same..! How is assessed the fact that a prediction is conceptually correct? Using types?
* The bottom part of Table 2 is surprising. How come for the task of predicting Head, the model trained only at predicting heads (GEN(t,r => h)) performs worse than the model trained only at predicting tails (GEN(h,r => t))?

---

### Official Review · AnonReviewer2 · 2017-11-30
**Review Genealized Graph Embedding Models**

**Rating:** 3
**Confidence:** 4

**Review:**

The paper proposes a new method to compute embeddings of multirelational graphs. In particular, the paper proposes so-called E-Cells and R-Cells to answer queries of the form (h,r,?), (?,r,t), and(h,?,t). The proposed method (GEN), is evaluated on standard datasets for link prediction as well as datasets for node classification.

The paper tackles an interesting problem, as learning from graphs via embedding methods has become increasingly important. The experimental results of the proposed model, especially for the node classification tasks, look promising. Unfortunately, the paper makes a number of claims which are not justified or seem to result from misconceptions about related methods. For instance, the abstract labels prior work as "ad hoc solutions" and claims to propose a principled approach. However, I do not see how the proposed method is a more principled than previously proposed methods. For instance, methods such as RESCAL, TransE, HolE or ComplEx can be motivated as compositional models that reflect the compositional structure of relational data. Furthermore, RESCAL-like models can be linked to prior research in cognitive science on relational memory [3]. HolE explicitly motivates its modeling through its relation to models for associative memory.

Furthermore, due to their compositional nature, these model are all able to answer the queries considered in the paper (i.e, (h,r,?), (h,?,t), (?,r,t)) and are implicitly trained to do so. The HolE paper discusses this for instance when relating the model to associative memory. For RESCAL, [4] shows how even more complicated queries involving logical connectives and quantification can be answered. It is therefore not clear how to proposed method improves over these models.

With regard to the evaluation: It is nice that the authors provided an evaluation which compares to several SOTA methods. However, it is unclear under which setting these results where obtained. In particular, how were the hyperparameter for each model chosen and which parameters ranges were considered in the grid search. Appendix B.2 in the supplementary seems to specify the parameter setting for GEN, but it is unclear whether the same parameters where chosen for the competing models and whether they were trained with similar methods (e.g., dropout, learning rate decay etc.). The big difference in performance of HolE and ComplEx is also surprising, as they are essentially the same model (e.g. see [1,2]). It is therefore not clear to me which conclusions we can draw from the reported numbers.

Further comments:
- p.3: The statement "This is the actual way we humans learn the meaning of concepts expressed by a statement" requires justification
- p.4: The authors state that the model is trained unsupervised, but eq. 10 clearly uses supervised information in form of labels.
- p.4: In 3.1, E-cells are responsible to answer queries of the form (h,r,?) and (?, r, t), while Section 3.2 says E-Cells are used to answer (h, ?, t). I assume in the later case, the task is actually to answer (h,r,?)?
- p.2: Making a closed-world assumption is quite problematic in this context, especially when taking a principled approach. Many graphs such as Freebase are very incomplete and make an explicit open-world assumption.
- The paper uses a unusual definition of one-shot/multi-shot learning, which makes it confusing to read at first. The authors might consider using different terms to improve readability.
- Paper would benefit if the model is presented earlier. GEN Cells are defined only in Section 3.2, but the model is discussed earlier. Reversing the order might improve presentation.

[1] K. Hayashi et al: "On the Equivalence of Holographic and Complex Embeddings for Link Prediction", 2017
[2] T.Trouillon et al: "Complex and holographic embeddings of knowledge graphs: a comparison", 2017
[3] G. Halford et al: "Processing capacity defined by relational complexity: Implications for comparative, developmental, and cognitive psychology", 1998.
[4] D. Krompaß et al: "Querying factorized probabilistic triple databases", 2014

---

### Decision · Program_Chairs · 2018-01-29
**ICLR 2018 Conference Acceptance Decision**

**Decision:**

Reject

**Comment:**

This paper does not meet the acceptance bar this year, and thus I must recommend it for rejection.